# Tailoring Perinatal Health Communication: Centering the Voices of Mothers at Risk for Maternal Mortality and Morbidity

**DOI:** 10.3390/ijerph20010186

**Published:** 2022-12-23

**Authors:** McClain Sampson, Wen Xu, Sahana Prabhu

**Affiliations:** 1Graduate College of Social Work, University of Houston, Houston, TX 77024, USA; 2Dell Medical School, University of Texas, Austin, TX 78712, USA

**Keywords:** mothers, home visit, recruitment, outreach, health communications, health promotion

## Abstract

The United States has the highest maternal mortality rate of any industrialized country. According to the Centers for Disease Control, Black women die at 2–3 times the rate of white women, and the infant mortality rate in the U.S. is 2.5 times higher than their White counterparts. Maternal and child health programs, such as Healthy Start, are an important gateway to increasing awareness, education, and referral to perinatal care and mental health services. This paper explored mothers’ perceptions of the importance of health and healthcare during pregnancy and postpartum and their preferences for communication from a community-based service program, such as Healthy Start. Data were collected from four focus groups with 29 expectant or current mothers. Most participants (57.7%) identify as Black or African American. They age from 24 to 43 with a mean of 31.7. We analyzed the data using the thematic analysis approach. Themes that emerged supported an overall desire for inclusive, strength-based educational materials. Use of advocacy-based health educational materials, materials that show diverse and realistic images of mothers, peer-based education through testimonials, and health education materials that are easy to understand and apply to one’s own experience emerged as the broad theme from the focus groups.

## 1. Introduction

The United States has the highest maternal mortality rate [1] and ranks 33rd out of 36 Organization for Economic Co-operation and Development (OECD) countries for infant mortality [2]. Vast racial disparities exist in maternal and infant birth outcomes. Black women die at 2–3 times the rate of white women, and the Black infant mortality rate in the U.S. is 2.5 times higher than White infants [3]. Two of the strongest correlates to adverse outcomes in maternal mortality, infant mortality, and severe maternal morbidity is late access to prenatal care and lack of follow-up in the postpartum period [4]. In general, low-income and minority women are less likely to initiate perinatal (the period of pregnancy through one year postpartum) care on time [5,6,7]. Reasons cited for late entry (past first trimester) to prenatal care among low-income women are public insurance, young age, low education, drug, and alcohol usage, or residing in rural communities [8]. Barriers to prenatal care in the first trimester among women of color include racial discrimination [7] and perceived low quality of care in patient-provider interactions [9]. Postpartum visits provide opportunities for intervention on risk factors for maternal mortality, such as hypertension, yet, low to no prenatal care is also associated with low rates of postpartum care [5]. 

Home-visiting programs, such as Healthy Start (HS), targeting childbearing-age women and families can serve as an important gateway to increasing awareness, education, and as a referral source to perinatal care and mental health services. First founded in 1991, HS is a federally funded, community-focused initiative that aims to “improve health outcomes before, during, and after pregnancy; and, to reduce racial and ethnic disparities in rates of infant death as well as negative health outcomes in the first 18 months of life” [10]. Currently, there are more than 100 HS sites in the U.S. serving communities with infant mortality rates at least 1.5 times higher than the national average. HS provides care coordination that addresses various social determinants of health of our participants, such as health insurance, food, transportation, housing, mental health, etc [11]. Using an evidence-based curriculum, HS also provides home visiting services to mothers and caregivers to increase their health management behaviors [11]. The service frequencies are determined after screening the women’s risks for adverse maternal and infant health outcomes. In this particular HS site, we provide weekly encounters for women at high risk, who might be, for instance, a mother who is having prenatal complications but is not seeing a provider. For women with medium and low risks, we provide bi-weekly or monthly service.

Although these programs serve “hard to reach” families, community-based home visiting programs still face low recruitment and retention challenges. A study of 274 UK women who were either pregnant or having babies under eight weeks and were eligible for a home visit service found that only 40.6% (n = 112) agreed to receive the support [12]. A more recent study with 2409 US mothers in a home visit program reported that 39.2% (n = 944) mothers stayed in the program for less than three months, and only every one in four (n = 591, 24.6%) women made it to longer than a year [13]. Even when the services were delivered at home, certain families were still reluctant to participate [13]. Socially, educationally, and economically disadvantaged women and women who live in more under-resourced neighborhoods are less likely to accept home-visit or wraparound services [12], and they also tend to be less actively engaged [13].

When faced with the challenges of low recruitment and retention in home visit services, especially among women from disadvantaged backgrounds, one possible solution is to adopt better strategies in reaching and communicating with the target population. Prior studies have discussed strategies to improve recruitment in intervention research settings, but few discussed communication strategies to increase recruitment for service programs. Based on their experience in a behavioral intervention trial targeting low-income overweight pregnant women in the San Francisco Bay area, Coleman-Phox et al. [14] concluded that the in-person approach and outreach at hospital-based prenatal clinics yielded the highest recruitment rates. A study among pregnant women in Canada [15] compared the traditional recruitment methods (word of mouth, posters, local health news) with paid Facebook advertisements and found the latter more cost-effective. Another study promoting healthy weight gain among pregnant women supported using digital marketing (e.g., Google and Facebook advertisements) to deliver health education [16]. In summary, these studies primarily focused on the medium of communication (e.g., via what channels), without much attention to the messages themselves (e.g., the contents on the posters or the Facebook advertisements), and found that the ***way*** we communicate health messages matters. 

The onset of a global pandemic and restrictions to in-person gatherings posed more challenges in recruitment for community-based health service programs. The barriers to recruitment among health-focused social service programs such as Healthy Start threatened to further marginalize mothers already at risk for adverse pregnancy and postpartum outcomes. Closures of schools and daycares, due to the COVID-19 pandemic, put an even higher than usual burden on mothers leading to upward spikes in postpartum depression and anxiety [17]. 

This paper describes one Healthy Start site’s effort to learn how to better reach mothers in need. We used a qualitative approach to ask the following research questions: How do mothers perceive the importance of health and healthcare during pregnancy and postpartum? How would women like to be talked to or advertised to when it comes to a program focused on perinatal health, such as HS? This exploratory qualitative study used a descriptive approach [18]. A descriptive qualitative design seeks to discover the “who, what and where of experiences” (p. 338), does not rely on a particular epistemology and is appropriate for focus groups where a broad range of information about a shared phenomenon is desired [18]. This study aimed to deepen our understanding of mothers’ experiences with perinatal health and learn how community-based health programs can tailor their communication materials to resonate with women. Findings are reported congruent with how questions were asked in that themes reflect what women said about their experience and what they would like to see differently with communication. This style of theme is consistent with a descriptive qualitative design [18].

## 2. Methods

### 2.1. Recruitment of Participant

The inclusion criteria were: (1) women with at least one living child or pregnant, (2) living in the City of X (anonymous for review purposes), and (3) fluent in English. Mothers do not have to be in any service program to participate. We spread recruitment messages via social media (Instagram and Facebook), by word-of-mouth, and through partnerships with HS and other local family service agencies. Participants expressed their interest by filling out an online recruitment form (via googleform.com), indicating their availability, and sharing their demographic and pregnancy-experience-related information. Then, we reached out to the participants to confirm the interview time and assigned a unique participant I.D. (for note-taking purposes). 

### 2.2. Data Collection 

The data were collected through four focus group interviews, each with about 6 to 8 participants, and lasted about one hour. All interviews were conducted via Zoom. We used a breakout room to greet women and verify that they had signed the informed consent and been assigned a unique I.D. to maintain confidentiality. Women were then sent to the main zoom room to be greeted and interviewed by a facilitator who is also an HS outreach staff but was not involved in data analysis. The facilitator discussed ground rules for maintaining confidentiality. Following the focus groups, all participants were emailed a $25 Amazon gift card to compensate them for their time. To minimize the social desirability bias that may occur due to compensation for time, we used transparency in explaining the project at the beginning of each focus group and decided on an appropriate, not exorbitant, amount of money [19]. The amount of $25 is determined based on the average hourly wage of $27.78 in this area [20]. The University of X (anonymous for review purposes) Institutional Review Board approved this study. 

The facilitator did a quick ice-breaker activity before asking questions from the protocol. Participants usually offered their thoughts in turns. The researchers were also in the Zoom room, listening, asking follow-up questions, and answering inquiries about the research. The primary interview questions asked were: If you were designing a flyer for other mothers about pregnancy health, what words would you use? What words would you not use? What images would you relate to? Would you use in any pictures or feature any other visuals?How would you talk to another mother if you wanted to convey the importance of having regular visits with your doctor/midwife during pregnancy and after?Please review these existing education materials from our team. Are there any changes you would make to this to better engage and reach mothers?

We consulted with a professor of communication, health literacy focus, to obtain feedback on their focus group questions.

### 2.3. Analysis

We adopted the thematic analysis approach Braun and Clark suggested to analyze the audio data and notes collected from focus groups [21]. First, the research team familiarized themselves with the data by listening to the recordings and reading our notes, writing down our initial thoughts separately. Then, each author generated codes individually by writing the codes down and collating direct quotes to relevant codes across the data. Third, the authors compared their coding notes and combined them into themes. Fourth, we reviewed the themes again to check if they matched the codes and the original data. After we all agreed on the themes, we finalized the definitions and names for each theme.

## 3. Results

### 3.1. Participants’ Demographics

A total of 29 mothers participated in this study, and 27 completed the demographics survey before the focus groups. More than half of them (n = 15, 55.56%) identified as Black or African American. Their ages ranged from 24 to 43, with a mean of 31.7. About 14 women were not previously enrolled in the HS program. Five out of the 27 participants were pregnant at the time of the focus groups. Regarding their utilization of pre or postnatal doctor visits, less than half (n = 11, 40.7%) women had prenatal and postpartum care. Among 22 non-pregnant mothers, six had received prenatal care only, and four had received postnatal care only. One-third of the sample (n = 9) reported health complications during pregnancy. Out of the 22 non-pregnant, six were at one year or less postpartum. 

Overall, the participants expressed a preference for educational materials with advocacy-based language, diverse and realistic representations of mothers, peer to peer education, encouragement to prioritize mom, and materials that are easy to read and are online. 

### 3.2. Advocacy-Based Messaging Feels Empowering

Women spoke of wanting to feel understood and involved in their treatment plans with case managers and/or with their health care providers. For example, suggestions for language on outreach materials such as “we advocate for your health” and “your opinion and your needs matter”. Participants said they would be drawn to program materials that convey that women’s concerns and preferences will be treated as worthwhile. For example, women wanted to be able to express their opinions, concerns, values while receiving healthcare services. They also expressed a desire, yet hesitancy, to trust medical and social service providers. One woman suggested that program materials and interactions with providers in the program should convey “we are all in this together”. Mothers stated they wanted someone to “be in their corner”, to advocate for them and teach them how to advocate for themselves. 

Participants said that communication materials should clarify if the program can help navigate interactions with medical providers. For example, mothers would like to see materials from case managers such as a pamphlet: “Questions to Ask Your OB/GYN”. During these conversations, participants often mentioned health care providers being the first point of interaction that influences how they view their pregnancy health. Several women stated that they knew they should be asking case managers and doctors questions but did not know what to ask. One woman said: “I came to find out [later, after childbirth] that as soon as I was having contractions. I should’ve gone to hospital”. Another woman said: “I never knew the questions to ask. My husband is a paramedic, and even he didn’t know the questions to ask! If you don’t know the questions, you aren’t going to get the time from them”.

### 3.3. Give Mothers Permission to Prioritize Their Health

A theme emerged that mothers struggle to prioritize their own health, and this may create challenges when trying to engage mothers. A mother of a 3-year-old said, “there is a lot of focus on physical health and not mental health” when referring to societal messages. One participant suggested putting the sentence: “caring for myself is caring for my baby” on a poster or pamphlet. Having a sense of ownership was important to mothers. As one participant said, [we want to hear] “you are the mother, and you can choose”. One woman suggested, “Put something on the flier telling moms to slow down. Too many moms are working through pregnancy and on their feet twenty-four-seven. They die because they don’t have people helping them while they are pregnant. Tell moms to rest”. 

One participant suggested motivating women by appealing to their desire to care for the baby “[the] health of the child is reliant on the health of mom, so encouraging her care to herself as a way to care for baby, even postpartum would be good”. Another mother with an infant and a toddler said, “they [moms] will feel terrible to do anything for themselves so give them something for the baby”. Some participants used the words “self-care”, and one mother articulated the importance of loving oneself as a form of self-care, “Moms should be told how important self-love is. You have to remind yourself that. How the mom feels affects how the baby feels. You need that time to yourself”. 

### 3.4. Promote Inclusivity with Diverse and Realistic Images of Mothers

Most participants talked about the importance of materials to recognize diverse pregnancy and motherhood experiences, such as single mothers, mothers who identify as LGBTQ+, or mothers having a rainbow baby (a term for a child born to a family that has previously lost a child due to miscarriage, stillbirth or death during infancy). Participants also would like to see images of mothers from different racial, socioeconomic and educational backgrounds and in various types of relationships in any visual materials. Two women pointed out that program materials with pictures of a mother and father together, or the word “husband” would initially seem irrelevant to them because they are lesbians. They said: “not assuming who [is] your family. I’m married to a woman, and some people don’t have a partner. Referring to your boyfriend or your husband would turn me off. Just focus on the mom”. 

One Black mother said, “I would relate to images of other black women because of the disproportionate amount of us who struggle and even die through pregnancy and childbirth”. Another woman suggested, “involve mothers of all varieties of race, income, and education. Include everyone at the table when asking what moms need”. Using realistic portrayals of mothers is also very important. Several participants said that pregnancy and motherhood are not always glamorous, and they wanted to see authentic images of mothers that they could relate. One postpartum woman said: 

The guys [husbands] need to understand that’s not the real picture — and they need to have more patience. Some of the posters make it seem like it’s gonna be amazing. Have an image that looks like me with no makeup and tired.

### 3.5. Mothers Want to Hear from Other Mothers

Participants stated that they want to see their own experiences reflected by others who have been through pregnancy and postpartum. To help foster information delivery that encourages mothers to seek preventive care, women in our focus groups reported that they would like to see real-life experiences of other mothers and care providers through clips or testimonies. As one participant said, “I like to hear other women’s stories, it makes it relatable.” 

Hearing and relating to other mothers’ stories helped motivate women to make decisions on their own health care. A mother of 4 said, “Whether it is a mother’s first or fifth pregnancy, ya’ll should not assume they [mom] have all the information”. Mothers also expressed the need to have social media groups or chat sessions to share their experiences and learn from each other. A pregnant woman shared, “your program should do like a small mommy group. Like the one I have on Facebook. It’s anonymous. They can express themselves without being judged”. 

Some women shared that by hearing personal stories of women who had died or had pregnancy complications, they felt the dangers applied more directly to them versus just hearing it from providers. One woman said, “I saw other women go through rough stuff and it made me realize I should go”. In response to the question, “what would you tell your friend if you wanted her to go to the doctor during pregnancy or well-woman visit”, one woman answered, “I would tell my friends what could happen if she doesn’t go. I would offer to help with her kids so she could go”. One participant said, “it’s amazing how many moms are tired of hearing what their own moms say to them”. There is a need and a desire to connect and learn from peers. 

### 3.6. Recruitment Materials Must Be Easy, Quick-to-Read and Have Online Presence

Participants asked for easy-to-understand, quick-to-read materials for recruitment purposes. For example, one woman said “a lot of young moms don’t really know what postpartum is. I would see a flyer or brochure and I would be like “what is that”?” they didn’t know the term baby blues. A postpartum woman in the same group said, “Agreed”. I was also a young inexperienced mom and did not know what postpartum depression was and, was ashamed of my feelings towards motherhood at the time”.

Moreover, the materials should use words and visuals that are easy to understand and apply to mothers’ own experiences. One woman in the group said “we usually only see the mom’s baby bump or the inside of the uterus”. When discussing what might help women to know that Healthy Start could make the pregnancy and postpartum journey easier and healthy, one woman suggested that community programs should make it obvious that they will help women understand healthcare and healthcare insurance. When mothers do receive important health related information it may come at a time when the woman isn’t ready. A mother of a 3-month-old referenced when she was being discharged from the hospital after giving birth, “The info I received was a bag with a bunch of information. When I got out (of the hospital), they gave me the presentation, but I really wasn’t paying attention. I was ready to get up out of there”. 

In terms of language, it was appreciated that this Healthy Start site’s recruitment materials were in English and Spanish. Materials should avoid medical or academic wording and instead utilize “everyday lingo”. For example, using “connection” to replace “referrals”. The design should also emphasize words that would quickly catch mothers’ attention, such as “free”, “confidential”, “community”, “empowerment”, “advocacy”, “awareness”, “safe space”, “wellness”, “self-care”, “support”, and “trust”. In terms of visual design, the materials should use more vibrant and inviting colors and avoid intimidating colors such as red, black, and other dark ones.

## 4. Discussion

In this study, women who were either current or potential clients in a local Healthy Start program were asked questions related to their experiences of health and their perception of the need for perinatal care. In addition, they were asked specifically about messaging in relation to the local Healthy Start program to improve outreach and education materials. The overriding theme we heard from our participants is that mothers wanted to be communicated with in a way that empowers and educates them, normalizes the various experiences of pregnancy, and they want to connect with other mothers as sources of education and support. Below, we discuss how the five themes are relevant to crafting marketing messaging as well as programmatic recommendations.

We discovered that mothers across all groups expressed a desire for agency and the ability to advocate for themselves. The term self-advocacy is used in various disciplines and settings and can be applicable to our findings. The term is used to mean that a consumer of health-related services is able to voice their personal preferences for care, get their needs met and feel some sense of control over their lives [22]. Social service programs such as Healthy Start can help bridge gaps in provider-patient interactions and strive to bolster a mother’s self-advocacy skills. There is evidence to support that nurses or social workers in community family services programs are able to act as health literacy coaches and increase the understanding of the health information provided [23,24]. Case managers should be aware that women of color are likely to feel that their concerns during pregnancy and birth are dismissed or denied by healthcare providers [25]. This awareness should inform how a case manager empathizes with a mother’s potential resistance to seeking prenatal and postnatal care.

Moms need permission to prioritize themselves, but urging them to do so may prove challenging. An example of how prioritizing one’s health for mothers may evoke mixed feelings is seen in Barkin and Wisner’s research [26]. They reported two conflicting themes that emerged during interviews with mothers regarding the valuation of self-care: self-care is critically important, but on the other hand, selflessness is necessary to be a good mother. The latter theme is anecdotally evident in this Healthy Start site as case managers note that participants schedule and attend their baby’s well-child visit but are inconsistent in their own care, L Keys, K Tompkins, M Homes, and Y Hills (personal communication, 4 May 2021) reported they have observed this pattern happened across their clients in a Healthy Start team meetings. Case managers at this local HS site also noted many participants address their baby’s need for health insurance before their own.

Representation matters for mothers in need of support. Inclusivity should be promoted with diverse images and representations of moms and families. Participants shared that if they were to see images of mothers and families that do not look like them, they would be less likely to view the program information on social media or in pamphlets. Healthy Start programs target communities that have at least 1.5 times higher the infant mortality rate than the national average [10]. Non-Hispanic Black (10.8 deaths per 1000), Native Hawaiian or Pacific Islander (9.4 out of 1000), and American Indian (8.2 out of 1000) infants are at the highest risk for dying before their first birthday [27]. Recruitment materials will differ depending on local demographics of risk, but overall, women of different races and ethnicities must be represented. 

Diverse representation in partnership and family structure was also desired by our participants. Among all births in the United States, 40% are among unmarried women [28]. A 2016 report from the National Center for Health Statistics revealed favorable attitudes for living together, but unmarried, when raising children. Importantly, 81.6% of women ages 25–34 said that it is acceptable for an unmarried woman to raise a child. This same report showed an increasing trend of favorable attitudes toward gay and lesbian childrearing [29]. It is likely that potential audiences for perinatal health communication messaging are hungry for the representation of a variety of examples that reflects modern parenting and family. 

The theme of moms wanting to learn from other moms is critical to consider in marketing and programmatic planning. Prior research purports that mothers view other mothers as valuable sources of information [30] and support [31,32]. Healthy Start sites should consider hosting peer-to-peer groups or mentoring programs alongside the typical case management strategy. Our finding is supported in existing research. [33] meta-synthesis found that first-time parents want early and realistic information related to pregnancy and birth, and they desire peer-to-peer education. Furthermore, there was a discussion on the need for education aimed at fathers. As Health Start sites are tasked with educating fathers on their role in perinatal health [10], the peer-to-peer format could benefit this gap as well. 

Recruitment materials that are quick to read and have an online presence emerged as a theme. Here, again, case managers or health educators in the Healthy Start program have an opportunity to act as translators or brokers of important health information that may have been given but not well received in the hospital. When choosing marketing materials, program administration should consider how the materials will land with an audience that may share the commonality of pregnancy but also represent a range of awareness and health literacy regarding perinatal health. The outreach materials need to provide easy-to-understand information and not make assumptions that women already know terms like postpartum depression and health complications. 

Previous qualitative studies support our findings. A qualitative study by Smith et al. [34] with mothers, clinicians, and community-based organizations explored their perspectives on reducing preterm birth. Similarly to our findings on mothers want advocacy, they concluded that Healthcare and social service providers should focus on whole-person care by acknowledging experiences of racism and using inclusive and culturally diverse approaches that engender respectful care and support for Black women, who bear the brunt of high rates of maternal morbidity [34]. Another study by Heaman et al. [35] with prenatal care providers (nurses, midwives, obstetricians, and family physicians) explored barriers and facilitators to prenatal care among inner-city women. Related to our findings, these providers suggested investment in relationships with mothers through rapport building and by making the clinic environment welcoming to diverse clientele [35]. Moreover, they also suggested that public awareness campaigns were needed to convey the importance of prenatal care among populations who face multiple barriers and lack relatable information [35]. 

Recognizing the time required to go through all the contents, designers should decrease the number of words to avoid overwhelming the readers. However, they could include links or barcodes to direct mothers who would like to have more information. Programs could also consider using popular social media platforms like TikTok and YouTube to increase awareness of the importance of quality and early prenatal care and follow-up. 

## 5. Limitations

This study has several limitations. First, the study was conducted during the COVID-19 pandemic, thus, we did recruitment and interviews entirely online, possibly leaving out mothers who did not have access to the Internet. Second, since this Healthy Start site was launched in the spring of 2019 and recruitment was slowed during 2020–2021, we did not have a large pool of participants to recruit from and thus chose not to limit focus group participation to only Healthy Start participants. We found it a little ironic that for a study to understand why it was hard to reach mothers for home visit programs, we had a hard time recruiting participants ourselves -- which again proved how important it is to discuss communication in recruitment. To reach an ideal sample size, we decided to expand our inclusion criteria to women who had not taken part in a home visiting program at the time of the interview but were eligible for the service. Third, the program was based in a metropolitan area in Southwest U.S., and our findings might not be transferable to programs based in different settings, such as rural areas. 

### Recommendations for Future Research

As family health research began to advocate for the importance of father involvement [36], we recommend future research to incorporate the women’s partners’ perspectives in studying maternal health communication. For home visiting programs like HS, encouraging prolonged engagement is a common challenge. A study with the Nurse Family Partnership, a similar maternal and infant home visiting program, reported a high attrition rate of 49.5% from enrollment to when the child reached 12 months old [37]. When people refuse to participate in the service or withdraw shortly after enrolling into the service, usually they just do not respond to calls and texts anymore. We recommend future researchers to investigate factors associated with low engagement and high attrition. 

## 6. Conclusions

Community-based health promotion programs can mitigate maternal morbidity and mortality risk through person-centered care and early home-based intervention. Yet, they are often plagued with recruitment issues. This study interviewed current and potential program participants to learn about their perspectives on communication during service recruitment. Our findings might be transferable to other community-based health promotion programs located in similar urban settings that seek to improve the recruitment of mothers at risk.

## Data Availability

Please contact the corresponding author.

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
