# Peer review of "Tailoring Perinatal Health Communication: Centering the Voices of Mothers at Risk for Maternal Mortality and Morbidity"

_ijerph, 2022, doi:10.3390/ijerph20010186_

Round 1

Reviewer 1 Report

COMMENTS TO THE AUTHORS of the paper presented under the title " Tailoring Perinatal Health Communication: Centering the Voices of Mothers at Risk for Maternal Mortality and Morbidity

Reference. Manuscript ID: ijerph-1975183

I think that the theme it addresses is important: the objective of this scientific article is: Lines 12-14

This paper explored mothers’ perceptions of the importance of health and healthcare during pregnancy and postpartum and their preferences for communication from a community-based service program, such as Healthy Start

To make myself understood, I expose in red and italics the literal text exposed in the article. I write in black, and in normal font, my appreciations.

1.- Data collection, section 2.2.

1.1. The data were collected through four focus group interviews which each had about

6 to 8 participants and lasted about one hour. All interviews were conducted via Zoom.

We used a breakout room to greet women and verify that they had signed the informed

consent and been assigned a unique I.D. to maintain confidentiality. Women were then

sent to the main zoom room to be greeted and interviewed by a facilitator who is also an

HS outreach staff but was not involved in data analysis. The facilitator discussed ground

rules for maintaining confidentiality.

Was it enough to address the three points indicated at the end of this section in one hour? Who considered that a discussion forum of 6-8 participants would be enough in one hour? Today the videos, at least one, could be made available to people interested in researching this topic. The issue of confidentiality could be resolved by presenting only the audio. If this is not presented, it is impossible to believe these results.

1.2.- Following the focus groups, all participants were emailed a $25 Amazon gift card to compensate them for their time.

The authors should question the influence that the fact of having paid for it may have had on the participation of the sample. There is a lot of literature on the matter that questions the veracity of the answers in this sense. They should at least reference it.

1.3. The University of X (anonymous for review purposes) Institutional Review Board approved this study.

This makes no sense. The University that consented to carry out this study should be named. This is a fundamental requirement for doing any work with people. There must be an identifiable ethics committee that allows the conduct of any study

1.4.- The facilitator did a quick ice-breaker activity before asking questions from the protocol. Participants usually offered their thoughts in turns. The researchers were also in the Zoom room, listening, asking follow-up questions, and answering inquiries about the research. Primary interview questions asked were:

1. If you were designing a flyer for other mothers about pregnancy health, what  words would you use? What words would you not use? What images would you relate to? Would you use in any pictures or feature any other visuals?

2. How would you talk to another mother if you wanted to convey the importance of having regular visits with your doctor/midwife during pregnancy and after?

3. Please review these existing education materials from our team. Are there any changes you would make to this to better engage and reach mothers?

We consulted with a professor of communication, health literacy focus, to obtain feedback on their focus group questions.

In one hour it is absolutely impossible for 6-8 people to respond thoughtfully to the content of the three questions presented at the end of this section. The authors should report the audios.

2.- Analysis, section 2.3

2.1.- We adopted the thematic analysis approach Braun and Clark suggested to analyze the audio data and notes collected from focus groups [17]. First, the research team familiarized themselves with the data by listening to the recordings and reading our notes, writing down our initial thoughts separately. Then, each author generated codes individually by writing the codes down and collating direct quotes to relevant codes across the data. Third, the authors compared their coding notes and combined them into themes.

Are the research team and the authors the same people? The authors in the complementary material should show these notes and these codes. If not, it is impossible to get an idea of how in just one hour, they were able to resolve these issues with each of the Zoom groups.

2.2. Fourth, we reviewed the themes again to check if they matched the codes and the original data. After we all agreed on the themes, we finalized the definitions and names for each theme

Again, are the research team and the authors the same people? What does it mean then, we reviewed?

3.- In the method section, the authors do not define the type of study or the design carried out. They should say that it is exploratory, descriptive research, which could be closer to a case study due to the small incidental size of the sample (N=29), and in which a study of the content of the answers has been carried out in a qualitative study.

4.-In the Results section, section 3, the authors show the results in 6 points, the following:

3.2 Advocacy-based messaging feels empowering

3.3 Give Mothers Permission to Prioritize their Health

3.4 Promote Inclusivity with Diverse and Realistic Images of Mothers

3.5 Mothers want to hear from other mothers

3.6 Recruitment materials must be easy, quick to read and have online presence

And in each of these 6 subsections, show some responses from the participants. In addition, the answers shown in each of these subsections are presented as segmented into themes, or nuances that have been appreciated. I think that they should show the responses of the subjects in 6 tables, and in each of the tables, highlight the different nuances of each of them. In this way, it would be easier to understand all this content analysis, and also how the authors have organized it.

5.- This work is already old because the pandemic is no longer there. The answers of the subjects who responded now may be different. Is it not possible to add someone interview conducted after the pandemic?

6.-It would also be good to have the impression of the couples of these women. What do their partners think? Do they think their wives are neglected in these aspects? I don't know if there is any study about it, but at least a comment should be made about it.

7.- Is there any research in this sense, where similar questions have been asked, or similar, or the same, to any health group, for example, nurses, doctors, or people who work in health centers? There should be some comments about it.

Author Response

Please see the file attached

Reviewer 2 Report

This study explored mothers’ perceptions of the importance of health and healthcare during pregnancy and postpartum and their preferences for communication from a community-based service program. However, a few comments are as follows:

1. In the Abstract and Introduction, the authors demonstrated that Black women die at 2-3 times the rate of white women and the Black infant mortality rate in the U.S. is 2.5 times higher ran White infant. In this study, Most participants (57.7%) identify as Black or African American. Is there any difference between the Black and White in this current study?

2. In the Introduction section (Line 41-45), the authors described this home-visiting programs (HS project), however, it is not clear what kinds of services the project provides. For example, how often does the program provide services for the parturient women? What forms and contents are included? What is the difference from the services provided by the hospital or what services or support can the hospital provide?

3. In the Introduction section (Lines 31-37), it was introduced that some low-income people could not have the prenatal examination on time or could not enjoy the medical and health services during pregnancy due to some social factors such as economic and educational level, which led to the high incidence rate and mortality of these people during pregnancy and childbirth. In this case, how can the HS project help them? Is it because the services provided by HS cannot meet the needs of these recruitment targets, so they cannot be well recruited? In this case, can we simply change the recruitment strategy or communication method to improve the current recruitment difficulties?

4. In the Methods section (Lin 95-101), the authors introduced the recruitment methods of the interviewees in this study and discussed how to better recruit those "hard to reach" mothers. In this case, whether there is bias in recruiting participants through the Internet? The interviewees who may be recruited usually pay more attention to these information or participate in these social projects. In this case, how can we better understand the ideas of those who refuse to participate in the HS project or withdraw after a short time of participation? In Line 110-111, does the additional economic compensation of participants easily lead to bias in sample selection?

5. In the Results and Discussion section (Line 141). The authors mentioned that the age distribution of the participants in this study is 24-43 years old, and the span is relatively large. Do people of different ages and with different numbers of children have different needs for communication when conducting data analysis? In addition, this study included both pregnant women and postpartum women, do different groups have different views in the group interview? Appropriate discussions can be conducted on these contents.

Author Response

Please see the file attached
